# Value-added scores show limited stability over time in primary school

**Valentin Emslander**[1]*, **Jessica Levy**[1], **Ronny Scherer**[2], **Antoine Fischbach**[1]

**1** Luxembourg Centre for Educational Testing (LUCET) at the University of Luxembourg, Faculty of Humanities, Education and Social Sciences, University of Luxembourg, Esch-sur-Alzette, Luxembourg,
**2** Centre for Educational Measurement at the University of Oslo (CEMO), Faculty of Educational Sciences, University of Oslo, Oslo, Norway

* valentin.emslander@uni.lu

**Data Availability Statement:** We uploaded our Minimal Dataset and Codebook to a public repository here (https://doi.org/10.17605/OSF.IO/VWD73).

## Abstract

Value-added (VA) models are used for accountability purposes and quantify the value a teacher or a school adds to their students' achievement. If VA scores lack stability over time and vary across outcome domains (e.g., mathematics and language learning), their use for high-stakes decision making is in question and could have detrimental real-life implications: teachers could lose their jobs, or a school might receive less funding. However, school-level stability over time and variation across domains have rarely been studied together. In the present study, we examined the stability of VA scores over time for mathematics and language learning, drawing on representative, large-scale, and longitudinal data from two cohorts of standardized achievement tests in Luxembourg ($N$ = 7,016 students in 151 schools). We found that only 34–38% of the schools showed stable VA scores over time with moderate rank correlations of VA scores from 2017 to 2019 of $r$ = .34 for mathematics and $r$ = .37 for language learning. Although they showed insufficient stability over time for high-stakes decision making, school VA scores could be employed to identify teaching or school practices that are genuinely effective—especially in heterogeneous student populations.

## Introduction

Can the effectiveness of a teacher or a school be quantified with a single number? Researchers in the field of value-added (VA) models may make this exact argument [1, 2]. VA models are used to calculate a score that represents the learning gains a student has received through their teacher or school. The VA score quantifies the difference between the expected achievement of students with similar background characteristics and their actual achievement [3]. Positive VA scores signify higher-than-expected achievement, given the student's background characteristics (e.g. socioeconomic status [SES], language, or prior achievement), whereas negative scores imply lower-than-expected achievement. Attempting to make a fair comparison between schools, these student VA scores can be averaged per school (or teacher) and indicate the value a school adds to its students [4, 5] independent of their

**Funding:** The present research was supported by the Observatoire National de la Qualité Scolaire (https://onqs.lu/de/) within the "Project Systematic Identification of High Value-Added in Educational Contexts (SIVA)" and a grant to VE from the Doctoral School in Humanities and Social Sciences of the University of Luxembourg (https://wwwen. uni.lu/fhse/doctoral_school). The funders had no role in study design, data analysis, decision to publish, or preparation of the manuscript.

**Competing interests:** The authors have declared that no competing interests exist.

background. Fig 1 illustrates such a comparison for one school with a high VA score and one school with a low score. In this example, the students from the two schools had comparable starting characteristics (e.g., prior achievement) and were thus expected to perform similarly from a statistical perspective. However, the students from School A performed better than what was statistically expected for a comparable group of students, indicating that this school offered *added value* to its students' achievements. School A would thus receive a high VA score, whereas School B would receive a low VA score.

Because this approach seems reasonable for quantifying school effectiveness and teacher effectiveness, VA modeling was adopted for accountability purposes in U.S. schools in the late 90s, and its application has since resulted in a surge in high-stakes decision making in educational settings [6]. In such VA-based assessment systems (e.g., Education Value-Added Assessment System, EVAAS), changes in the VA scores of teachers or schools are used to reward highly effective teachers, by offering them a tenured position, and schools, by allocating more funding. At the same time, teachers with low VA scores might face in extremis unemployment, and schools with low VA scores might need to operate on a tighter budget [7, 8]. This allocation of resources is based on the notion that differences in a student's VA score from one year to another are due solely to this student's teacher or school.

Parallel to the rise in applications of VA models, research interest in evaluating the actual performance and precision of VA models has bloomed [6, 9]. Due to a lack of consensus on how to calculate VA scores, and because VA scores differ greatly in their accuracy depending on the exact model used to calculate them [6, 9, 10], a school's VA score could differ from one year to the next. Hence, a school with a high VA score one year may receive a low VA score the next year without any actual change happening at that school. VA scores may change because of variation in teaching or school effectiveness but also due to different

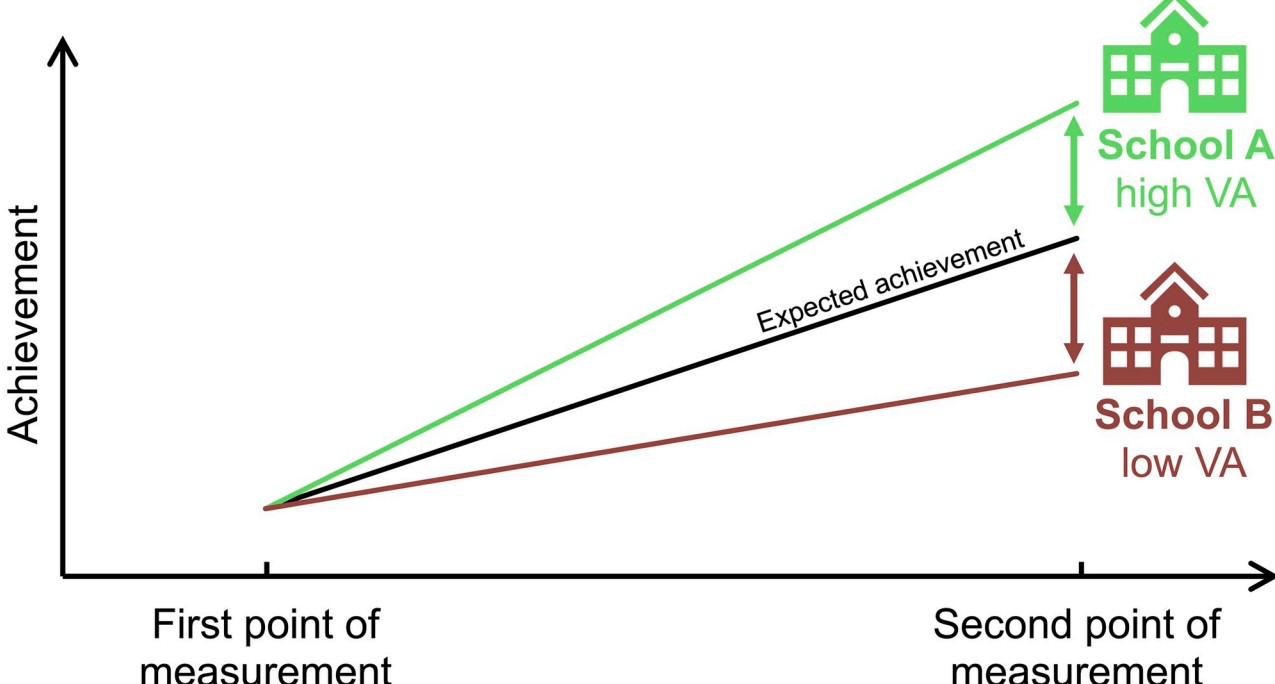

**Fig 1. Illustration of high and low VA schools performing above and below what is expected of them from one measurement point to another.** Double-headed arrows signify the VA score as the difference between a school's expected achievement and its actual achievement.

errors in the measurement and construction of VA scores [11]. Due to this instability, researchers and policymakers have argued that VA scores are not suitable for high-stakes decision making [12, 13], and their use should be restricted to informing teachers and schools about how they can improve their schools. To utilize VA scores for high-stakes decision making (e.g., teachers' tenure or allocation of funding), these scores need to be highly stable over time. However, findings on this school-level stability is mixed, with some studies indicating generally high stability in school VA scores over time [14, 15] and others reporting instability [e.g., 13, 16].

Furthermore, there is no consensus on which independent or dependent variables should be used in VA models beyond prior achievement [9, 17], and the stability of VA scores could vary between different outcome domains (e.g., between language and mathematics). If VA scores lack stability *over time*, we cannot be sure whether these changes are due to actual changes in teachers and schools or measurement issues, such as error or choice of models. If VA scores lack stability *across outcome domains*, differences could be driven by different teacher and school practices in the two domains but also by variation in the measurement of the two domains so that they cannot be meaningfully compared. If VA scores lack stability *over time and across outcome domains*, we would not be able to attribute VA score variation to changes in effectiveness over time or to educational differences between domains. Thus, the use of VA scores as the primary information for high-stakes decision making is in question, and the inferences drawn from them could be compromised. To shed light on the "stability problems" of VA models [6], we estimated VA scores for two cohorts of students from 151 primary schools and examined the stability of these scores over time and across the most frequently used outcome domains: mathematics and language learning. In our large-scale data set, we included all eligible primary schools in Luxembourg and thus worked with population data. We chose to investigate students at the beginning of their school careers because younger students have been found to show greater response to interventions than older students [18]. Our study extends the body of research on VA score stability by considering multiple domains and primary schools, adding a rich set of background variables to our analysis, and using state-of-the-art VA models and covariate combinations [10, 19, 20].

## Theoretical framework

### Value-added models and their use

Effectiveness is often difficult to compare across schools because no two schools are alike in the composition of students' language background, SES, or prior achievement. To solve this difficulty in comparing schools, researchers have drawn on VA scores that control for student composition factors (e.g., students' language background, SES, and prior achievement) and single out the "net effect" of school effectiveness [21]. VA scores can be calculated in different ways in terms of variables and choice of statistical models [9, 10, 22]. However, the underlying idea, which originated from economics [23], is the same for all of these statistical models: When controlling for all available background variables and prior achievement, all gains in achievement that are left are likely to be due to teacher or school effectiveness (for an in-depth literature review of research on teacher and school VA scores, please see [9, 17, 24].

The idea underlying VA scores can be expressed in two simple statistical steps: Eq 1 shows the first step, in which the expected achievement $\hat{y}$ is estimated for every student $i$ in school $j$ as a function $f$ of their initial characteristics $x_{ij}$ at an earlier time point (e.g., prior achievement). This function $f$ is usually a linear regression or a multilevel model [9, 25].

$$\hat{y}_{ij} = f\left(x_{ij}\right)$$

(1)

Eq 2 shows the second step, in which a VA score is estimated for each school *j* by calculating the mean difference (i.e., residuals) between the expected achievement $\hat{y}$ and the actual achievement *y* for all *n* students in this specific school *j*. This is equal to the average error term *e* of all students *i* in school *j*.

$$VA_j = \frac{\sum_i^j (y_{ij} - \hat{y}_{ij})}{n_{ij}} = \frac{\sum_i^j e_{ij}}{n_{ij}} \tag{2}$$

A high VA score indicates that students in school *j* achieved above what was expected of them, as was the case in the example of School A in Fig 1. A low VA score indicates that students in school *j* achieved below what was expected statistically. The idea is to find the "true" school effect—namely, the value a school adds to its students' achievement—by statistically controlling for everything that cannot be changed by a school. Following this idea, everything that is left can be seen as the true school effect (i.e., the residuals; Eq 2). Therefore, it is important to ensure a high level of quality in the initial prediction step (i.e., Eq 1).

Researchers have explored different uses of VA models to render issues of educational effectiveness visible. One use is to employ VA models for high-stakes decisions, such as decisions about teachers' salaries or tenure or a school's funding [26]. In the US, where VA research is flourishing [9, 17], VA models are applied to policymaking in large parts of the country [25, 27]. In other parts of the world, VA models are used as well, for example, in Italy, Portugal, Brazil, the United Kingdom, and the Netherlands [16, 28–30]. Another practice is to use VA models to identify high-performing schools and the factors that determine their success [31].

The use of VA scores for accountability purposes is based on highly debated scientific findings and a mixed research literature. While using VA scores is a significant improvement over using only achievement tests to compare school effectiveness [16], researchers still debate crucial aspects of VA scores. Some issues of VA models have already been identified and tackled, such as creating a consensus on how to best estimate VA scores [9, 10, 17]. The most widely used models are the multilevel and linear regression models, with the former outperforming the latter [19, 20]. However, the stability of VA scores over time in primary schools still needs to be examined to ensure that VA scores are informative for this educational level.

## Stability of value-added scores over time

Let us consider one of this year's top-performing primary schools: Do we expect this school to perform as well next year and in two years? We would probably expect this school's VA scores to be stable over time because we can assume that a school's VA scores are susceptible to outside events only to a small extent.

Research on school-level stability over time is still scarce and has produced mixed results. Table 1 gives an overview of prior research on the stability of school VA scores over time with the included variables and samples. On the one hand, some studies have supported the stability of VA scores over time: Ferrão [14], for example, found a moderate level of stability in VA scores over two years in Portugal with 65% of scores remaining in the same quartile of VA rankings. She recommended using VA scores for school improvement, especially in countries with high retention rates. Similarly, Thomas et al. [15] investigated changes in VA scores of English secondary schools over ten years. They found that only one school (which had a low VA score) out of 16 schools was able to meaningfully raise its VA score across a period of more than four consecutive years, whereas most other schools' VA scores improved only over two consecutive years before stagnating or declining again. On the other hand, several researchers found a lack of VA score stability in their data. In their study of all secondary

**Table 1. Overview of prior research on the stability of school value-added scores over time, their included variables, samples, and conclusions.**

| Reference | Included Variables | Sample | Findings & Conclusion |
|---|---|---|---|
| Ferrão (2012) | • Achievement in mathematics or reading and prior achievement<br>• Student characteristics: SES, gender, self-declaration of race/skin color, kindergarten attendance, SEN, attendance of mixed class, grade repetition<br>• School characteristics: composition SES, type of school governance | • Portugal: 45 primary and 14 elementary and lower secondary schools with two cohorts spanning grades 1 to 8 were researched over two years each<br>• Over 4 years (2005–2008) | • Moderate level of stability over two consecutive years with 65% of scores remaining in the same quartile of VA rankings.<br>• Use VA scores for school improvement, especially in countries with high retention rates |
| Gorard et al. (2013) | • Achievement in English, mathematics, and science<br>• Student characteristics: gender, SEN, ethnicity, free school meals, first language, school changes, age, IC, IDACI<br>• School characteristics: Variance of student achievement within a school | • England: All secondary schools with VA scores ($n = 2,897$)<br>• Over 5 years (2006–2010)<br>• High VA: confidence interval does not include the mean VA score | • No school showed consistently high VA scores across five consecutive years<br>• VA scores are unstable over time and should not be used in practice<br>• The missing data problem must be solved |
| Thomas et al. (2007) | • General Certificate of Secondary Education points of students, prior achievement at age 11 with the Cognitive Abilities Test<br>• Student characteristics: age, gender, ethnicity, free school meals, poverty, SEN, school changes<br>• School characteristics: none, as gender composition was about 50% in all schools | • England: Ten consecutive cohorts of 16-year-old secondary school students ($n = 134$) in one large school district<br>• Over 10 years (1993–2002) | • One out of 16 schools was able to meaningfully raise its VA score across a period of more than four consecutive years<br>• Most other schools' VA scores improved only over two consecutive years before stagnating or declining again<br>• Low VA schools are more likely to improve |
| Perry (2016) | • Achievement at ages 7 and 11, averaged for each cohort<br>• Student characteristics were aggregated on the school-level<br>• School characteristics: gender (%), SEN (%), English as an additional language (%), free school meals (%), child looked after status (%), number of students in a cohort, coverage (inclusion in the measure) | • England: nearly all primary and secondary state-schools<br>• Over 4 years (2011–2014) | • Moderate to large correlations between VA scores over one ($r = .59$-$.61$), two ($r = .45$-$.46$), and three years ($r = .35$) in primary schools<br>• Student characteristics could explain substantial variance in VA scores<br>• Avoid school VA scores as a basis for policymaking or other high-stakes decisions |

*Note.* $n$ = number of schools in the sample; SES = socio-economic status; SEN = Special Educational Needs; VA = Value-Added; IC = students who have been 'In Care' at any time while being at this school; IDACI = Income Deprivation Affecting Children Index measuring deprivation based on student postcode. Please see the respective original study for more details.

schools in England, Gorard et al. [13] found that none of the schools showed consistently high VA scores across five consecutive years. They interpreted their finding as evidence that VA scores were unstable over time and should not be used in practice until the reliability of the scores could be improved. Perry [16] found moderate to large correlations between VA scores over one ($r = .59$-$.61$), two ($r = .45$-$.46$), and three years ($r = .35$) and showed that student characteristics, such as English as an additional language and eligibility for free school meals, could explain 11% and 35% of the variance in VA scores in primary and secondary school, respectively. Despite this moderate to large stability but given the dependence on student characteristics, Perry [16] and more recent research from the UK [32] recommended avoiding school VA scores as a basis for policymaking or other high-stakes decisions.

The extant literature has discussed some reasons for the instability of VA scores. Perry [16] hypothesized that the most variance could be found within rather than between schools. The small amount of between-school variance might therefore not be very informative [33]. As noted above, Perry [16] also considered variables outside of the control of teachers and schools as potential reasons for instability. This is a common phenomenon in longitudinal

studies such that the actual change in the school hardly drives variation, but rather, the variability is driven by outside events, such as a successful team-building intervention or the admission of several new students. Changes within one cohort over multiple years could also simply be due to the maturation of the students, whose cognitive abilities [34] and peer relationships [35] develop rapidly before and during school. But teachers can also change such that professional development, positive feedback, and experience could have a positive influence on the VA scores of the entire school, whereas critical, personal life events, or additional responsibilities could have a negative impact [28].

Measurement error and regression to the mean might also be drivers of change in VA scores [16]. Regression to the mean is most prevalent in extreme groups (high and low VA scores), which might make these two groups most prone to variations. This effect was mostly ignored in VA research [36]. Looking at the extreme groups of schools with high and low VA scores, some of these schools might be in this group only due to the inevitable measurement error in achievement scores [37]. Such schools that were misplaced accidentally due to this measurement error would likely be closer to the mean at the next measurement point, implicating unwanted variance in VA scores over time [36].

A solution to the stability issues in VA scores might be, for example, to control for a rich set of background variables and to estimate the VA scores with the same models over time [6]. Model choice and included variables are highly influential in estimating VA scores [10, 16, 38] and should therefore be held constant when looking at the stability of VA scores.

## Different outcome domains

While there seems to be some consensus about which variables should be included when estimating VA scores [9], less is known about the impact of different outcome domains. Only a few studies have looked at stability across diverse measures of the same achievement domain [39] or even contrasted two different achievement domains [40]. At the student level, in accordance with the reciprocal internal/external frame of reference model [41], we might expect differences between mathematics and language achievement to be large. This model describes the link between, for example, a student's achievement in mathematics or language with their academic self-concept in the other domain in a longitudinal setting. However, this effect might average out when considering the teacher or school level on which the VA scores are summarized.

Comparing VA scores and students' regular achievement measures helps evaluate the VA scores' validity, which is crucial for the credible use of VA scores in policymaking or other kinds of high-stakes decision making. Naturally, policymakers would like schools with a high VA score to do well not only on mathematics tests but also on language tests and vice versa. Thus, a comparison of different outcome domains is indicated to investigate whether VA scores are equally meaningful for the domains of mathematics and language.

Prior research on both school and teacher VA scores—two closely related research areas—has focused on the comparison of correlations between VA scores with different outcome domains or different measures of the same outcome domain [8, 39, 40, 42]. On one end of the spectrum, Ferrão and Couto [40] found that school VA scores had strong correlations with students' mathematics and Portuguese performance as the outcome domains in Grades 2 through 5, ranging from $r = .43$ to $.70$. Looking at teacher VA scores, Sass [8] investigated their stability over time and across two test instruments and found a correlation of $r = .48$ between VA scores of two different achievement tests. He also found that this correlation was higher than the correlation of VA scores over time ($r = .27$). At the same time, he acknowledged the substantial difference between the two outcome measures, which he attributed to differences

in testing material, potential ceiling effects, and differences in pressure to perform, as one measure was a high-stakes test and one was a rather low-stakes test [8]. Lockwood et al. [42] found large correlational differences between two subscales from the same mathematics test on the teacher level. They attributed large parts of the variation in the VA scores to these different measures in a middle school sample, suggesting that VA scores are very sensitive to the choice of outcome measure. These results were later replicated by Papay [39], thus corroborating the conclusion that the choice of outcome measures can have a larger influence on the stability of VA scores than the model specifications. Going beyond Lockwood et al.'s [42] original study, Papay [39] also explored differences between three comprehensive reading achievement measures as outcome variables. He found correlations of $r = .15$ to .58 between VA scores with these three different outcome measures. These results indicate some comparability between the measures but not enough for high-stakes decisions. In a real-life situation, almost half of all teachers would have had different salaries if the outcome measures underlying their VA scores changed, as Papay [39] demonstrated. Taken together, these findings demonstrate the limited stability of VA scores across different outcome domains.

Focusing on the stability of VA scores between different tests of the same outcome domain, prior research has left one question open: What is the stability of VA scores between not only different outcome tests (e.g., two different mathematics subtests) but between different outcome domains (i.e., mathematics and language)? We would expect that a school with a high VA score is not only helping its students excel in one domain (e.g., mathematics) but is also facilitating learning in another domain (e.g., language) [39]. Stated differently, a VA score that shows great variability in its predictive power across different outcome domains would not be stable across domains and would thus not be helpful. Then again, VA scores that were well-aligned no matter whether mathematics or language was used as the outcome domain would provide an argument for the validity of school VA scores, especially if the VA scores in the two domains showed similar levels of stability.

## International use of value-added scores

In the US and Europe, VA scores have experienced increasing research interest [9, 23]. With the No Child Left Behind Act [43] and the Race to the Top Act [44], VA models have been applied for policymaking in large parts of the US [25, 27]. Parallel to their increase in use, VA scores have also experienced critical resistance from researchers and teachers due to methodological flaws and their real-life implications [12, 45]. After 2009, several unsuccessful lawsuits had been filed against the use of VA scores in decisions about teachers' remuneration or tenure, leaving many educators with smaller pay or without a contract at all after their VA-based assessment. In 2017, in the school district of Houston, TX in the US, however, the federal court ruled it unconstitutional to terminate a teacher's contract on the basis of undisclosed VA score data [46]. Overall, VA scores have found most of their use in the US, which has led to several lawsuits against their use in high-stakes decisions in education policy.

Parallel to the development of the "Tennessee Value-Added Assessment System" in the US (TVAAS [47]), for instance, the French ministry of education introduced VA scores to be used in school evaluations as well ("Indicateurs de valeur ajoutée" [48, 49]). As discussed above, VA scores are used in the United Kingdom, where other variables in addition to prior achievement are used to estimate the VA scores [16]. In this way, student characteristics such as ethnicity, SES, or gender could also inform VA scores and increase their fairness. Here, VA scores are usually used as an approximation of a school's effectiveness and to provide school performance rankings [13, 16]. In the Netherlands, VA scores are used to identify disadvantaged primary schools that are at risk of underperforming so that targeted interventions can be offered to

them [30]. Further, Ferrão [29] reviewed research on school-level VA scores in Brazil and Portugal.

In a nutshell, most countries outside the US avoid using VA scores for high-stakes decisions. Nonetheless, they apply VA scores in order to inform such decisions as one of multiple tools that can be used to estimate a facet of school effectiveness. Such examples introduce VA scores as a means of identifying high-performing teachers or schools to learn from them or their low-performing counterparts in order to support such lower performers in a less punitive and more appreciative way.

## Unsolved issues in value-added research and the Luxembourgish context

Prior research has presented several unsolved issues and open questions on the use of VA scores, for example: How stable are school VA scores over time? How stable are they across outcome domains? These questions are crucial for highly diverse educational contexts, because they can profit the most from reliable and valid VA scores. For example, diversity in the student population can arise from diverse languages spoken at home, a migration background, or a family's SES. This diversity leads to different preconditions for learning mathematics and new languages (or even the language of instruction) and thus shapes students' school careers [50] and school completion [51]. Consequently, VA scores improve by including relevant background information, such as the languages spoken at home, alongside prior achievement.

The Grand Duchy of Luxembourg provides one such highly diverse educational context with a multilingual student body, leading to gaps in students' achievement that widen with age [52]. This multilingualism is reflected in the fact that in 2020, only 43% of all students in Grades 1 and 3 spoke Luxembourgish or German at home [53]. Alongside students who come from a socioeconomically disadvantaged family or attend a secondary school in the lower tracks, students who do not speak the language of instruction are specifically challenged to do well in school [54].

Another specificity of the Luxembourgish primary schools is that they operate in four learning cycles, spanning two years each. Cycle 1 starts with two years of preschool before Cycle 2 spans the first and second years of school. These two-year cycles continue until Cycle 4 ends in Grade 6. Within one cycle, students typically have the same class teacher until they progress to the next cycle, where they get a new class teacher. Thus, longitudinal studies conducted in Luxembourg, such as the Luxembourg school monitoring programme [55], are usually conducted every other year to correspond to the structure of the learning cycles. In this way, similar results within one learning cycle, for example, due to having the same class teacher, are not overinterpreted.

In a context as diverse as Luxembourg, VA scores could provide a much fairer measure of school effectiveness than averaging the standardized achievement of all the students in one school. However, VA scores can only be used in these contexts if they exhibit a sufficient level of stability across time and across different outcome domains. In other words, they need to exhibit satisfactory reliability and validity. Otherwise, VA scores would fail to flexibly adjust to the constantly evolving language and school landscape [56] and should thereby not be used in high-stakes decision making.

## Relevance of value-added score stability

The relevance of the stability of VA scores is directly linked to their reliability and validity. If they are stable over time in multiple outcome domains (touching on their reliability and validity, respectively), they could be a great additional tool for evaluating school effectiveness. This would be especially helpful in highly diverse educational contexts, where controlling for

students' backgrounds makes the effectiveness estimate much more meaningful. This effect might be smaller in rather homogeneous contexts that lack variation in students' and schools' backgrounds.

If VA scores prove to be unstable over time and in different outcome domains, however, their use should be cautioned. A larger variety in VA scores that cannot be explained by the predictor variables (i.e., noise in the data) makes VA scores less reliable and less valid. They would be less reliable if the same school with unchanged circumstances places high one year and low in some other year without apparent real-life change occurring between these years. Similarly, VA scores would be less valid if they had vastly different results for different outcome domains (i.e., mathematics and language) without an actual difference in achievement, leading to a lack of real-life explanatory power. Therefore, without sufficient explanatory power to set apart high- and low-VA schools and without the stability to demonstrate a reliable estimation of school effectiveness, VA scores would not be suited for making high-stakes decisions.

In a concrete application example, at an unlucky school in the US whose VA score plummets due to measurement error or changing circumstances, several teachers might lose their jobs. Across the big pond, in the United Kingdom, a school with an unstable VA score might move to the top of the effectiveness ranking undeservedly, attracting more students without having improved its school climate or bettered its general level of instructional quality. Such unwarranted fluctuations need to be considered, as their real-life implications can be radical and, in some cases, harmful. To avoid disadvantageous outcomes from the application of VA scores, they need to either (a) show sufficient stability across time and outcome domains (touching on their validity and reliability) or (b) be abolished as a means of measuring a school's effectiveness for high-stakes decisions and should be used only for informative purposes.

## The present study

With the present study, we examine the stability of school VA scores over time and quantify differences in stability between two outcome domains (i.e., mathematics and language achievement). Specifically, we address the following research question: *How stable are school VA scores over time and across outcome domains*?

To answer this research question, we first calculated VA scores at the school level, ranked all the schools accordingly, and estimated the correlations of the ranks over time to arrive at a stability estimate. To calculate the VA scores, we used the same selection of covariates across the two outcome domains. To choose the covariates, we reviewed models of school learning [e.g., 57, 58], prerequisites, and correlates of students' achievement as well as recent findings on the selection of covariates in school VA models [20]. We followed the approach specified by Levy et al. [20], who found that including prior mathematics achievement, prior language achievement, and covariates related to students' sociodemographic and sociocultural backgrounds (i.e., socioeconomic status of the parents, languages spoken at home, migration status, and sex) in multilevel school VA models could help leverage between-school differences in student intake and in the resulting school VA scores. By examining the stability of school VA scores and comparing different outcome domains, we seek to contribute to the existing debate on the stability and use of VA scores in educational effectiveness.

## Method

### Participants

The present study drew on longitudinal large-scale data from the Luxembourg School Monitoring Programme *Épreuves Standardisées (ÉpStan)* [55]. The *ÉpStan* assesses all students in

Grades 1, 3, 5, 7, and 9 in Luxembourg at the beginning of every school year. By doing so, every student who follows the usual path through school is tested every other year. Data on the students are collected in three main areas: academic competencies (in mathematics and languages), motivation and emotion, and background variables (e.g., language background and SES). In the present study, we used data from the cohorts of Grade 1 students in the years 2015 and 2017 to inform our VA scores for the Grade 3 students in the years 2017 and 2019 and form a large longitudinal data set with school VA scores at two time points (2017 and 2019).

The final data set comprised $N$ = 7,016 students, nested in 151 primary schools in Luxembourg. Students were included if they participated in Grade 1 and two years later in Grade 3. We thus excluded data from students who (a) had missed data collection in Grade 3 (e.g., due to sickness or grade repetition) or (b) were enrolled at a different school in Grade 3.

The *ÉpStan* received approval from the national committee for data protection and has a proper legal basis. Current ethical standards were respected at all times [59]. The participating students and their parents or legal guardians were duly informed before the data were collected and had the opportunity to opt out. All data were pseudonymized with a so-called "Trusted Third Party" (for more information, see [55]), in accordance with the European General Data Protection Regulation, to ensure the privacy of students and their families. In the present study, we used an anonymized data set.

## Measures

To calculate the VA scores, we included measures of prior academic achievement and measures of an outcome domain (i.e., mathematics achievement and language achievement) all with high psychometric quality. To further inform the VA scores, we also included measures of sociodemographic and sociocultural background variables.

**Academic achievement.** Academic achievement plays two crucial roles in VA modeling: Whereas prior academic achievement is used as a covariate, current academic achievement is irreplaceable as an outcome variable. We used measures of mathematics and language achievement for Grade 1 simultaneously to calculate all VA scores. For the outcome measures in Grade 3, however, we report all results separately, once for mathematics and once for language achievement. Our choice to incorporate both mathematics and language achievement into our analyses was further corroborated by meta-analytic evidence of their mutual relationship [for a recent meta-analysis on the mutual relationship between language and mathematics, see, 60].

In the first months of Grades 1 and 3, students' mathematics and language achievement scores were collected with standardized achievement tests. Expert groups consisting of teachers, content specialists in teaching and learning, and psychometricians developed these tests to ensure the content validity of these tests [61], based on the Luxembourgish national curriculum standards [62]. On the day of testing, the students completed the achievement tests in their own classrooms in a paper-and-pencil format. Most items were designed as closed questions and scaled by a unidimensional Rasch model [61, 63, 64]. Warm's Mean Weighted Likelihood Estimates were used (WLE [65]) to indicate student achievement. We used these WLE values and their standard errors to calculate the reliability of all the achievement scores in the R package TAM version 3.3.10 [66]. Table 2 shows these reliability coefficients for both mathematics and language in Grades 1 and 3 in 2015–2017 and 2017–2019.

**For mathematics achievement, students completed a test in** Grade 1 in Luxembourgish because the language of instruction in preschool is Luxembourgish. Whereas Luxembourgish can be described as a language cognate to German [67], politically and culturally it is a language of its own. The students answered items from three domains of mathematics competence: "numbers and operations," "space and shape," and "size and measurement" (see, for a

**Table 2. Reliability coefficients for the achievement scores for the 2015–2017 and 2017–2019 data sets.**

| Variable | 2015–2017 | 2017–2019 |
|---|---|---|
| Mathematics achievement in Grade 1 | .84 | .85 |
| Language achievement in Grade 1 | .71 | .73 |
| Mathematics achievement in Grade 3 | .93 | .93 |
| Language achievement in Grade 3 | .83 | .83 |

more comprehensive explanation https://epstan.lu/en/assessed-competences-21/). In Grade 3, the students took the mathematics tests in German because the students had been taught in German during Grades 1 and 2. Again, the students answered items from three mathematics competence domains: "numbers and operations", "space and form", and the novel area of "quantities and measures" (see, for a more comprehensive explanation https://epstan.lu/en/assessed-competences-31/ [68]).

**For language achievement, students completed a test in** Grade 1 in Luxembourgish because the language of instruction in preschool is Luxembourgish. The standardized language tests consisted of "listening comprehension" and "early literacy comprehension." To assess listening comprehension in the two language competence domains "identifying and applying information presented in a text" and "construing information and activating listening strategies," the students listened to different kinds of texts in an audio recording. For early literacy comprehension, the students were tested on three competence domains, namely, "phonological awareness," "visual discrimination," and "understanding of the alphabetic principle" (see https://epstan.lu/en/assessed-competences-21/). We averaged the scores for listening and reading comprehension as prior language achievement in the VA model to have one single score for language achievement.

In Grade 3, the students took the listening and reading comprehension language achievement tests in German, which had been the language of instruction during Grades 1 and 2. The listening comprehension test consisted of two domains of competence: "identifying and applying information presented in a text" and "construing information and activating listening strategies." For reading comprehension, the students were again tested on two competence domains, namely, "identifying and applying information presented in a text" and "construing information and activating reading strategies/techniques" (see https://epstan.lu/en/assessed-competences-31/ [68]). Analogous to prior language achievement in Grade 1, we calculated a mean score across listening and reading comprehension in the German language, resulting in a single dependent variable for language achievement.

In the present study, we conducted secondary analyses of archived data sets and had only limited information on the psychometric quality of the achievement tests. However, as these data sets formed the basis of political and practical decisions in Luxembourg, the psychometric quality of the tests had been optimized in that regard [61, 69]. More specifically, as mentioned above, several expert panels developed all the items for the domain-specific achievement tests to ensure their content validity. After pilot testing, all items underwent psychometric quality checks concerning their empirical fit to the Rasch model—that is, the model that was used to generate WLE estimates to represent students' domain-specific achievement in Grades 1 and 3. To additionally ensure adequate testing between student cohorts at the same grade level (e.g., between Grade 1 in 2017 and Grade 1 in 2019), all test items were examined for differential item functioning. These psychometric quality checks were complemented by analyses of convergent and discriminant validity. Finally, the students' domain-specific achievement scores, as represented by the WLE scores, demonstrated reliability coefficients between .70 and .90, which were considered sufficient [70].

**Table 3. Descriptive data of the two samples used in the present study.**

| Years of the sample | N | % of female students | % of students speaking Luxembourgish with at least one parent at home | Mean (SD) SES | Migration status (% native) |
|---|---|---|---|---|---|
| 2015–2017 | 3443 | 50% | 50% | 50.3 (15.6) | 48% |
| 2017–2019 | 3573 | 49% | 48% | 49.7 (16.0) | 47% |

*Note.* SES = Socioeconomic status as measured by HISEI (Highest International Socio-Economic Index of Occupational Status).

**Sociodemographic and sociocultural background variables.** All parents with a child in Grade 1 were asked to complete a questionnaire on their child's sociodemographic and sociocultural background. Table 3 shows the descriptive data for each of the two samples (2015–2017 and 2017–2019) and for the entire sample of 7,016 students. The sociodemographic and sociocultural distributions were similar in both data sets. Parents identified their occupation from a list of categories that were based on the ISCO (International Standard Classification of Occupations) classification. To approximate the parents' SES, an average value of all occupational categories was computed on the basis of the validated ISEI (International Socio-Economic Index of occupational status, see [71]) scale. For the Grade 1 sample, parents reported a mean ISEI value of 50.3 for the sample from 2015 and 49.7 for the sample from 2017. These mean values were only slightly above the average ISEI for all OECD countries of 48.8 from the first PISA tests in 2000 [72].

To indicate a child's migration status, parents further specified where they and their child were born. This was translated into three categories of migration status: "native," "first generation," and "second generation." We created a dummy variable for migration status with "native" as the reference category. In the 2015–2017 sample, 48% of the students had a "native" migration status, and 47% of the students in the 2017–2019 sample had this status. To complement their parents' answers to the questionnaire, the first graders also answered a questionnaire on the language(s) they spoke with their parents. Not speaking any Luxembourgish at home could be considered a disadvantage for students in Grade 1 because both the testing and the preschool instruction were in Luxembourgish. Therefore, we coded the language(s) spoken at home as a dummy variable to distinguish between students who spoke Luxembourgish with at least one parent (reference category) and those who did not speak any Luxembourgish at home. In the 2015–2017 sample, 50% of the students indicated that they spoke Luxembourgish with at least one parent, and 48% in the 2017–2019 sample did so. Furthermore, a data set from the Ministry of National Education, Students, and Youth provided information on the students' sex, with 50% (2015–2017) and 49% (2017–2019) girls in the samples.

## Data analysis

All analyses were performed in R version 3.6.1 (R Core Team, 2021). After preparing the data sets and imputing the missing data, we estimated the VA scores for the schools. Part of the data preparation and the estimation of school VA scores were analogous to the study by Levy et al. [20], who used different data from the school monitoring programme [55] from the years 2014 and 2016.

**Data preparation.** Because a criterion for inclusion in the study was that students participated in the Grade 3 achievement tests, there were no missing data in the achievement data in Grade 3. To impute missing data on the covariates, we used multiple multilevel imputation with 20 imputations, 50,000 burn-in iterations, and 5,000 iterations between imputations using the R packages *mitml* version 0.3–7 [73] and *jomo* version 2.6–9 [74]. The S1 and S2 Data show the R code for the data imputation for the data from 2015–2017 and 2017–2019, respectively.

**Estimation of school value-added scores.** We estimated the random effects within each school to obtain VA scores via Eqs 1 and 2 [37]. More specifically, these were the Level 1 residuals from the multilevel model, averaged within a school. In other words, all student-level VA scores were averaged into one VA score per school. To estimate the model, we used the log-likelihood as the estimator. We applied the *lmer* and *ranef* functions from the R package *lme4* [75] and defined the multilevel model as follows:

```
Achievement_in_Grade_3 ~ Prior_Math_Achievement + Prior_Langua-
ge_Achievement + SES + migration_status + language_spoken_at_home
+ sex + (1|school_ID)
```

In this model, the outcome variable represents either mathematics or language achievement in Grade 3. Prior achievement in mathematics and in language, SES, migration status, language spoken at home, and sex are covariates, thus being statistically accounted for.

To address our research question on the stability of VA scores across time and outcome domains, we ranked the schools by their VA scores from highest to lowest in one year. Then, we created three indicators of VA score stability. First, we estimated correlations between the two outcome domains for 2017 and 2019, respectively. Second, we checked for the stability of the school VA scores within one outcome domain over two years by calculating one correlation between the VA mathematics score in 2017 and the respective VA mathematics score in 2019, and we did the same for language. Third, we estimated the correlations between the VA mathematics scores in 2017 with the VA language scores in 2019, and vice versa (i.e., the correlation between language as an outcome domain in 2017 and mathematics as an outcome domain in 2019). Fig 2 graphically represents these three types of correlational indicators. The resultant correlation coefficients will be interpreted as indicators of VA score stability across time and domains [11, 39].

On the basis of these findings, we investigated the total number and percentage of schools that showed a stable or unstable VA score rank with mathematics as an outcome domain compared with language as an outcome domain. For this purpose, we consulted commonly used benchmarks [76] and decided to use four levels of VA scores to indicate a school's effectiveness:

- High VA scores are in the top 25% (highly effective schools)
- Upper medium VA scores are between the 50th and 75th percentiles (moderately to highly effective schools)
- Lower medium VA scores are between the 25th and 50th percentiles (moderately effective schools to schools that might need improvement)
- Low VA scores are in the lowest 25% (schools that need improvement)

We defined schools with a stable VA score as schools that remained in the same VA rank quartile in 2017 and 2019. We defined schools with an unstable VA score as schools that were in different quartiles in 2017 and 2019. The S3 Data shows the R code we used to analyze the correlations between the VA scores. The S1 Table shows the covariance table of VA scores with different outcome domains (mathematics and language) over time. The S1 Dataset shows the minimal dataset and codebook of school's VA quartile ranking across time and domains.

## Results

### Stability of value-added scores

We found positive correlations between school VA scores in both the mathematics and language outcome domains and across the two years of testing. Table 4 depicts the correlations for the school VA scores in the different outcome domains (mathematics and language) across

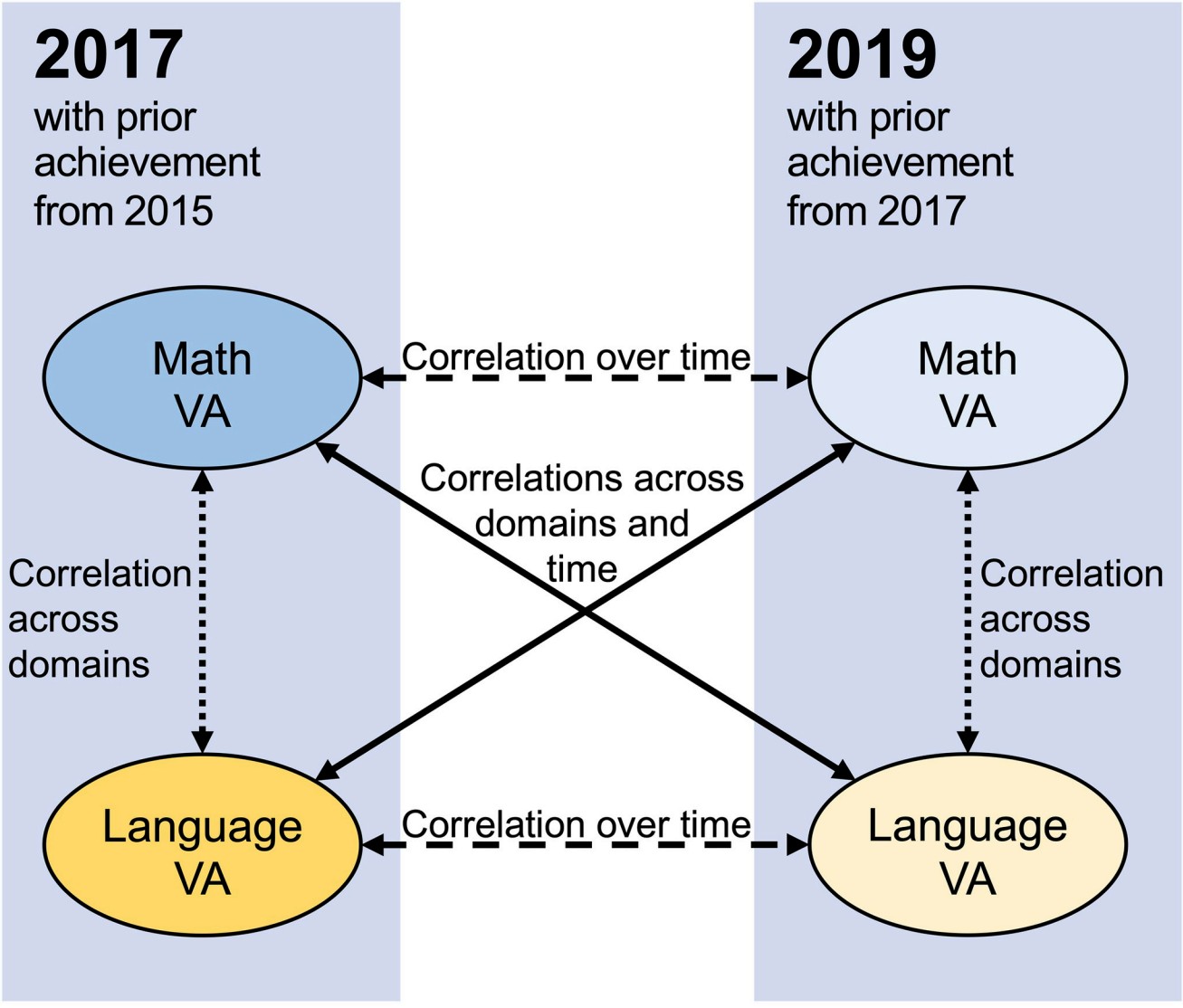

**Fig 2. Graphical representation of the VA score intercorrelations across time and domains.** Double-headed arrows signify the correlations we calculated.

**Table 4. Correlation table of VA scores in different outcome domains (mathematics and language) over time.**

|  | 2015–2017 Mathematics | 2015–2017 Language | 2017–2019 Mathematics | 2017–2019 Language |
|---|---|---|---|---|
| 2015–2017 Mathematics | - |  |  |  |
| 2015–2017 Language | .59 | - |  |  |
| 2017–2019 Mathematics | .34 | .22 | - |  |
| 2017–2019 Language | .32 | .37 | .47 | - |

*Note.* Correlations were calculated on the basis of $n$ = 7,016 elementary schools students' VA scores.

time. More specifically, the school VA scores in mathematics and language within the same years were moderately to highly correlated with correlation coefficients of $r = .59$ in 2017 and $r = .47$ in 2019. The correlations of the school VA scores within one outcome domain across the two years were smaller but still moderate. We found that the school VA mathematics scores from 2017 and 2019 were correlated at $r = .34$. Similarly, the school VA language scores showed a correlation of $r = .37$ across the two years. The correlations of the school VA scores across both the domains and time were smaller than the other correlations. The correlation between the school VA language scores in 2017 and the school VA mathematics scores two years later was $r = .22$. The correlation between the school VA mathematics scores in 2017 and the school VA language scores two years later was $r = .32$. Overall, we found moderate correlations across outcome domains but only small correlations over time in primary school for the school VA scores.

### Prevalence of schools with stable and unstable VA scores

Fig 3 shows a transition diagram illustrating the number of schools that changed or remained in their VA score rank quartile. Looking at the school VA mathematics scores, 54 out of 151 schools had the same rank in both 2017 and 2019. This was roughly one third of the schools with mathematics as an outcome domain that remained stable over the two years (approx. 35.8%). There were more schools in the highest and lowest ranks that remained stable (i.e., 34 schools) than in the two middle ranks (i.e., 20 schools). While most schools (i.e., 97 schools) changed one or two ranks up or down, 10 schools were classified as having a high VA score in one year and a low VA score two years later, or vice versa. When language was the outcome domain, the results were similar. We found that 63 out of 151 had a stable VA score over time. Again, most schools had a different rank in 2019 than the rank that was based on their VA score in 2017. Two schools had moved from the lowest to the highest rank, whereas 5 schools had gone from the highest to the lowest rank over the two years.

Overall, the results for mathematics and language were comparable. We also found that only about one third of the primary schools exhibited stable VA scores across the two years. The other two thirds fluctuated, with some schools changing their VA score rank position substantially.

## Discussion

VA models are used for accountability purposes in education and quantify the value a teacher or a school adds to their students' achievement. For this purpose, these models predict achievement over time and attempt to control for factors that cannot be influenced by schools or teachers (i.e., sociodemographic and sociocultural background). Following this logic, what is left must be due to differences in teachers or schools [5]. To contribute to the debate about the stability of VA scores over time and across outcome domains, we drew on representative longitudinal data from two cohorts of standardized achievement tests administered to a total of 7,016 students attending 151 primary schools in Luxembourg. First, we calculated correlations between the VA scores within and across time and outcome domains. Additionally, we investigated the total numbers and percentages of schools that showed a stable versus unstable VA score across time in mathematics and language as the outcome domains.

### Stability of value-added scores across time and domains

In our sample of primary schools, we found moderate correlations of $r = .34$ for mathematics and $r = .37$ for language as the outcome domains across a two-year period. These correlations are far from perfect (i.e., $r = 1$), moderate in size, and can thus be considered to show instability in VA scores over time. If VA scores worked perfectly, this instability could be explained by

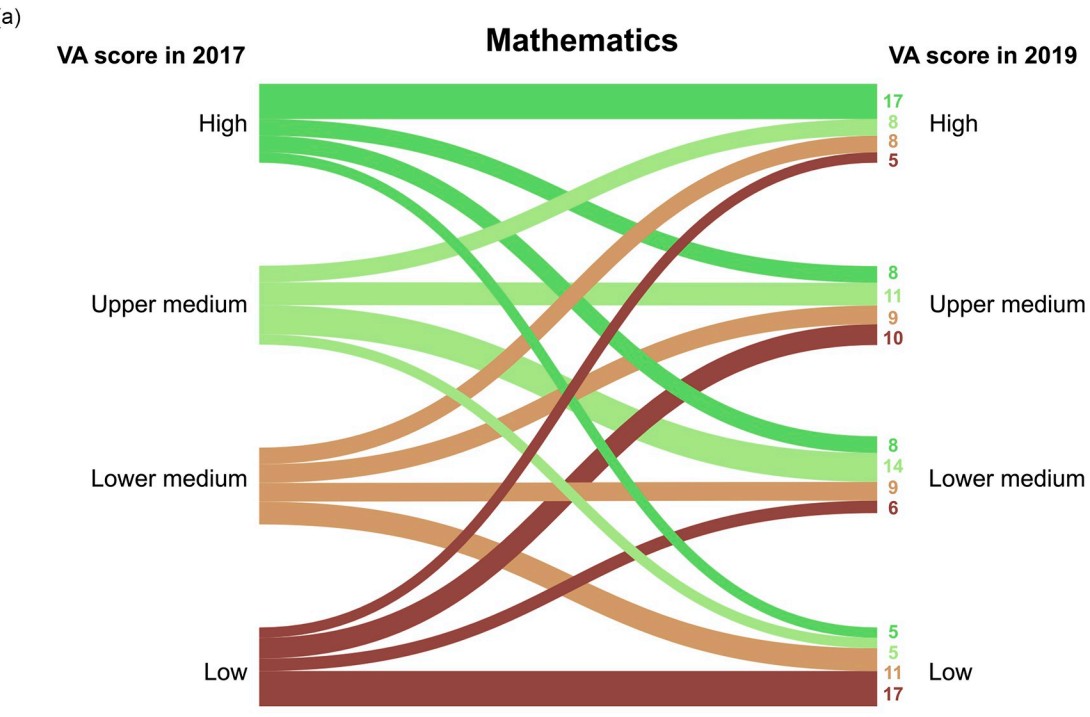

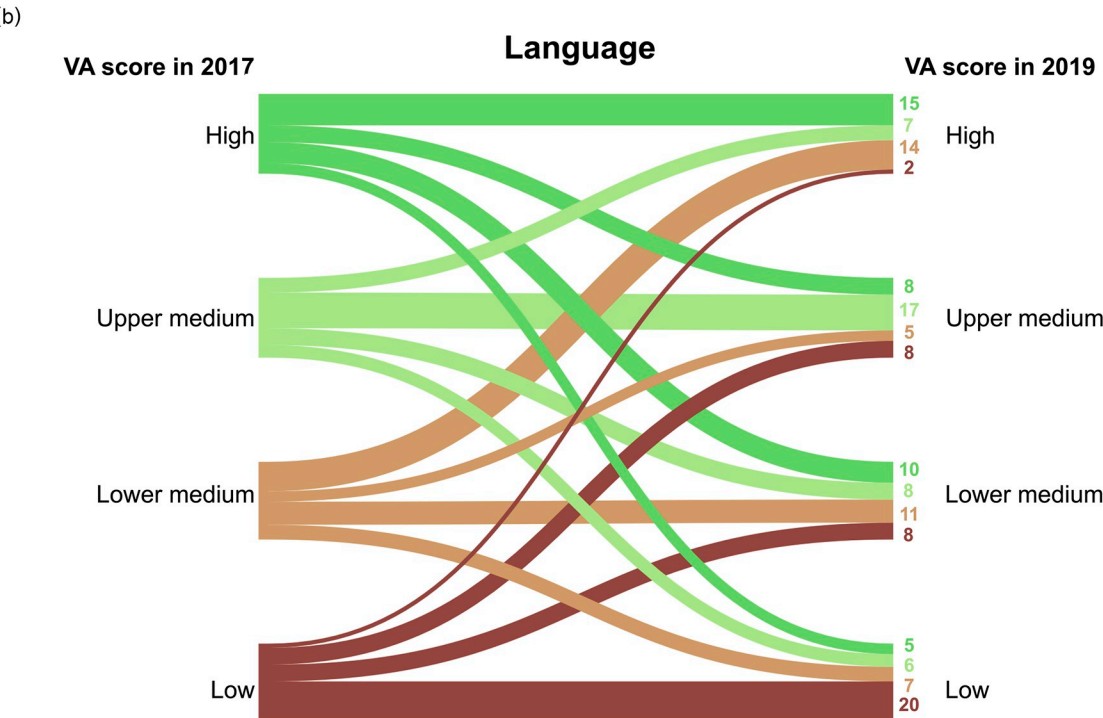

**Fig 3. Transition diagrams indicating changes in the schools' value-added score ranking quartiles from 2017 to 2019.** A school's position in the VA score ranking is indicated on either side of the transition diagram. High VA scores are in the top 25%, Upper medium VA scores are between the 50th and 75th percentiles, Lower medium VA scores are between the 25th and 50th percentiles, and low VA scores are in the lowest 25%.

actual improvement or decline in the schools' quality, but this is probably not the case because several other factors influence VA scores. Instead, researchers have pointed to the multiple sources of error and bias in VA scores introduced by, for example, variation on the student level and measurement issues [8, 13, 16]. Thus, the instability in VA scores we found should be attributed primarily to these disturbances rather than to changes in school effectiveness. The correlations we found across a two-year period in primary school were even somewhat smaller than those found in a secondary-school sample by Perry [16], who cautioned against the use of VA scores as a basis for high-stakes decision making and policymaking. We also share the interpretation expressed by Gorard et al. [13], who warned against the use of VA scores until consensus has been reached on how to handle missing data, which models to use, and until the predictive power of VA scores for school effectiveness has been unambiguously shown.

While we found large correlations across mathematics and language within the same years —2017 and 2019, respectively—we found only small correlations across domains and across years. Correlations were larger across outcome domains in the same year than they were for the same outcome domain across two years' time. Stated differently, time has a greater effect on VA stability than different outcome domains do. This could either indicate greater changes in school effectiveness between years than between domains or simply show less noise in data collection within the same year than over time due to similarity in assessments even in different domains. This interpretation is in line with research by Thomas et al. [15], who identified that time as a factor introduces considerable instability. The larger correlations within the same year across outcome domains replicate prior findings [8, 40].

The reasons for the instability in VA scores can be manifold. Of course, schools improving through effective leadership or successful teacher development introduce the kind of change and variation that is desired in VA scores [28]. As discussed above, however, school VA scores might vary for statistical reasons, such as greater variability within than between schools, measurement error, and regression to the mean. However, external variables might influence the stability of school VA as well, such as changes in a student cohort due to maturation, (adverse) life events of a class or an entire school, or changes in teachers or teacher behavior due to professional devolvement or personal circumstances, for example [8, 11, 16, 28]. However, most of these sources of variance are outside the control of the teachers and the schools but can introduce instability in VA scores.

Given our correlational findings and the multiple sources of instability, we conclude that VA scores have insufficient predictive power for the value a school adds to its students over time. Thus, these scores cannot be considered stable enough over time to be used for high-stakes decisions [13, 16]. As VA scores are a highly political topic and their use is controversially discussed, they should not be used as the sole indicator for a schools effectiveness on a public policy level [27, 77]. In the public interest, however, schools with stable VA scores can be used in research contexts for informative purposes, including learning from schools with stable high VA scores.

## Differences between outcome domains

We found that 34–38% of the schools showed stable VA scores from Grade 1 to Grade 3 by remaining in the same VA score ranking quartile in 2017 and 2019. These results were similar across outcome domains. Considering the similarity in the correlations between mathematics and language over time, the differences in the VA scores between the outcome domains can be considered small [78].

While 34–38% of the schools remained in the same VA score ranking quartile, all other schools had VA scores that changed quartiles between Grade 1 and Grade 3. As multiple

sources of variance may be influencing the stability of VA scores, the changes in the rankings of these schools might not be due to their actual improvement or deterioration but may have been driven by statistical or external factors. Thus, about two thirds of the schools were at risk of falling victim to unwarranted consequences. Especially for the 10 schools that changed from the highest to the lowest VA ranks or vice versa, this instability could have dramatic consequences if VA scores were applied for accountability reasons, such as decisions about funding or the closing of schools. With no major advances in the stability of VA scores by adding richer sets of background variables or more advanced statistical methods, on its own, "Value-added is [still] of little value" [79] to public educational policy.

While we did not find similar numbers of educational units that remained the same as Ferrão [14] did, we share her sentiment that VA scores could be used as indicators of school improvement. At the same time, we argue for the complementary use of VA scores, paired with observational, qualitative, and other kinds of data to obtain a broader picture of school effectiveness [31].

## Limitations and future directions

The present study has potential limitations. First, how to treat missing data is an important question in research in general, especially in VA research, as small changes in data handling might have large impacts and might sway real-life decisions. In the present study, we included only cases that had complete achievement test data in the two outcome domains and then imputed the other missing values as described above rather than using case-wise deletion methods. By excluding the incomplete achievement tests, we might have missed relevant cases, potentially introducing bias in the VA score. Future research should investigate the sensitivity of VA scores across different ways of handling missing data [20].

One specificity of the Luxembourgish school system is its structure in two-year learning cycles. During these two years, it is expected that the same teachers teach the students, as is customary in Italy for example [80]. Therefore, we did not investigate students in two consecutive years but in the first year of two consecutive learning cycles. In research, most VA scores are estimated from one year to the next. Had we focused on students in two consecutive years, however, with the same teachers, in the same classroom, and less time between the measurement points, VA score stability might have been higher, as the two measurement points would be more similar [16].

The two-year learning cycles can be prolonged by one year, so a child remains in the same learning cycle not for two but for three years, which is a form of repeating a grade. Retention is quite common in Luxembourg, leading to a large number of students participating in the *ÉpStan* in Grade 1 in 2017 but not in Grade 3 in 2019, for example. These excluded students tend to have a lower socioeconomic status, are likely to show lower achievement, and are less likely to speak the language of instruction with their parents than the included students. The practice of looking at students with regular educational pathways is in line with the common practice of estimating VA scores [81, 82], and VA scores can especially be used for constructive purposes in educational settings with high retention rates [14].

In the present study, we focused on primary school Grades 1 and 3 in Luxembourg, a highly diverse and multilingual school context compared with other countries. While our findings add further evidence of the instability of VA scores to the literature, we should not extrapolate our findings from Grades 1 and 3 to other grades or blindly take the results from Luxembourg and apply them to samples in vastly different educational settings. Perry [16] found that even in the same Grade 4 and Grade 6 cohorts, for example, VA scores had a correlation of only $r = .24$. And, as could be expected, cohorts in two consecutive years are more likely to be

similar than cohorts that are several years apart [16]. Further, VA scores were suggested to show greater variability between different cohorts than within one cohort [8, 80, 83].

Future research could tackle the stability of VA scores and the implications of stability in these scores in three distinct ways: (a) by investigating fundamental research questions on the parameters that influence the stability of VA scores, (b) by conducting a systematic review of both published and gray literature on the stability of VA scores over time, and (c) by looking at the real-life implications that (un)stable VA scores have for schools. To tackle open basic research questions, future research could extend the present study to replicate our findings in a less diverse and rather monolingual educational context. These are just a few directions future research could take, for which prior research has already provided some evidence. Findings from these endeavors could then be used to evaluate the specificity of the present study—in other words, whether Luxembourg, whose strength lies in a heterogeneous student population, is either significantly different from other countries or quite comparable to other places in the world.

Looking at the real-life implications for schools of (un)stable VA scores and their productive use outside of high-stakes decision making is another area that future research should explore. It will be informative to investigate differences between schools with stable high VA scores and those with stable low or moderate VA scores and learn about effective pedagogical strategies [14, 31]. Such a constructive use of VA scores can help create parsimonious samples, identify and appraise effective schooling, and aid schools that are in need of support. These could be future applications of VA scores and the present findings on VA score stability.

## Conclusion

The present study provided evidence for the moderate stability of primary schools' VA scores. Only 34–38% of the schools showed stable VA scores across two years with moderate correlations of $r = .34$ for mathematics and $r = .37$ for language achievement as the outcome domains. The number of stable schools did not differ greatly between mathematics and language. Real-life implications for schools may be consequential with only about one third of schools having a stable VA score over time. This finding indicates that VA scores should not be used as the only measure for purposes of accountability. Thus, both public and private educational services should refrain from using VA scores as the (sole) metric to rank schools in their effectiveness and policymakers need to consider the present controversy about school VA scores. Complementary sources of data to make appropriate educational decisions are strongly recommended, as school VA scores do not seem to be stable enough over time and theoretical assumptions about the VA scores don't seem to hold in practice [77, 84]. Nonetheless, we argue that VA models could be employed to find genuinely effective teaching or school practices—especially in heterogeneous student populations, such as Luxembourg, in which educational disparities are already an important topic in primary school [85]. Here, VA scores could help researchers look past these disparities and investigate the schools with stable positive VA scores and learn from them.

## Supporting information

**S1 Data. Data imputation for 2015–2017.**
(RMD)

**S2 Data. Data imputation for 2017–2019.**
(RMD)

**S3 Data. Correlations of value-added scores.**
(RMD)

**S1 Table. Covariance table of VA scores with different outcome domains (mathematics and language) over time.**
(DOCX)

**S1 Dataset. Minimal dataset and codebook of school's VA quartile ranking across time and domains.**
(XLSX)

## Acknowledgments

We would like to thank the national school monitoring team from the Luxembourg Centre for Educational Testing for providing access to the Épreuves Standardisées database. The authors wish to express their special thanks to Ulrich Keller, University of Luxembourg, for his insightful comments on the syntax of this project as well as to Professor Dr. Martin Brunner, University of Potsdam, for his valuable feedback on an earlier version of this manuscript.

## Author Contributions

**Conceptualization:** Valentin Emslander, Jessica Levy, Antoine Fischbach.

**Formal analysis:** Valentin Emslander, Jessica Levy.

**Funding acquisition:** Jessica Levy, Antoine Fischbach.

**Project administration:** Valentin Emslander.

**Resources:** Antoine Fischbach.

**Software:** Jessica Levy.

**Supervision:** Ronny Scherer, Antoine Fischbach.

**Visualization:** Valentin Emslander, Jessica Levy.

**Writing – original draft:** Valentin Emslander.

**Writing – review & editing:** Valentin Emslander, Jessica Levy, Ronny Scherer, Antoine Fischbach.

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
