## [Decision Letter · Decision Letter 0]

13 Oct 2022

PONE-D-22-22428Value-Added Scores Show Limited Stability over Time in Primary SchoolPLOS ONE

Dear Dr. Emslander,

Thank you for submitting your manuscript to PLOS ONE. After careful consideration, we feel that it has merit but does not fully meet PLOS ONE’s publication criteria as it currently stands. Therefore, we invite you to submit a revised version of the manuscript that addresses the points raised during the review process.

We look forward to receiving your revised manuscript.

Kind regards,

Pedro Ribeiro Mucharreira, Ph.D.

Academic Editor

PLOS ONE

Journal Requirements:

Reviewers' comments:

Reviewer's Responses to Questions

**Comments to the Author**

1. Is the manuscript technically sound, and do the data support the conclusions?

Reviewer #1: Yes

Reviewer #2: Partly

Reviewer #3: Yes

Reviewer #4: Yes

2. Has the statistical analysis been performed appropriately and rigorously? 

Reviewer #1: Yes

Reviewer #2: No

Reviewer #3: Yes

Reviewer #4: Yes

3. Have the authors made all data underlying the findings in their manuscript fully available?

Reviewer #1: Yes

Reviewer #2: No

Reviewer #3: Yes

Reviewer #4: Yes

4. Is the manuscript presented in an intelligible fashion and written in standard English?

Reviewer #1: Yes

Reviewer #2: No

Reviewer #3: Yes

Reviewer #4: Yes

5. Review Comments to the Author

Reviewer #1: The paper is a high-quality piece of research on the Value-added by a teacher or a school to their students’ achievement. It is based on a structured exploration of relevant literature, as well as on a rigorous empirical assessment, providing insightful results that are aligned with previous findings. Nevertheless, before publication, in the conclusions it is recommended to provide implications at the public policy level, that is, education policy including the supply and demand for both public and private education services.

Reviewer #2: The title "Value-Added Scores Show Limited Stability over Time in Primary School" addresses a topic of extreme interest, but difficult to read, with images that do not help to clarify the theory it presents.

The article with profound changes has potential for future publication, at the moment it seems unclear to me.

-Poor introdution

- Insufficient framing, I suggest updating the literature review and improving the specification of models. Presenting a table of the variables used in studies of the same genre and their respective conclusions, helps to better understand the framework and later the discussion and conclusion.

- Correct numeric values, for example, .15 means 0.15.

- Develop the method and improve the explanation of academic performance.

- Weak Data analysis, discussion and conclusion. It is unclear whether the data are used year by year, or pooled for the 3 years under study, namely in calculating the Covariance Table of VA Scores with Different Outcome Domains (Math and Language) over time.

Good luck for work development!

Reviewer #3: The statistical analysis seems to be very well integrated within the context, and the article is very thoroughly documented and well written. I recommend the article to be published as it is, and I consider it is a very useful contribution to the journal.

Reviewer #4: It seems to me, in my opinion, that the article focuses on a topic of great interest and relevance. It is reasonably well written, and provides a list of adequate references, both in number and current.

Maybe the images could be improved a little more, in terms of quality, but overall it seems like a good article.

6. PLOS authors have the option to publish the peer review history of their article (what does this mean?). If published, this will include your full peer review and any attached files.

Reviewer #1: **Yes: **João Leitão

Reviewer #2: No

Reviewer #3: No

Reviewer #4: No

---

## [Author Response · Author response to Decision Letter 0]

24 Nov 2022

PONE-D-22-22428: Details of revision for resubmission

PLOS ONE

Title: Value-Added Scores Show Limited Stability over Time in Primary School

November 2022

Dear Professor Ribeiro Mucharreira, dear Reviewers,

Before providing detailed responses to the reviewers’ comments and describing the changes we have made to the manuscript, we would like to take this opportunity to thank you for inviting us to resubmit this revised version. We thank the editor and the reviewers for their valuable input and insightful remarks. Overall, your comments and suggestions have led to substantial revi-sions of the manuscript, including the following:

• We added a more critical discussion on the public policy implications of our findings in the discussion and the conclusion. 

• We refined all three figures according to your recommendations to improve their quality and reworked Table S4 in the Supplemental Material.

• We further discussed and added recent literature and made sure to better explain our methods and the use of the longitudinal data.

We will now address each of the reviewers’ comments. Below, we fully reproduced the com-ments; our replies are depicted in italics, new or changed text passages in the manuscript are written in blue and underlined. We kindly refer you to the Reference section in the revised man-uscript for a full list of cited references. Finally, please accept our apologies for a certain amount of overlap between replies; we attempted to keep each reply exhaustive.

Yours sincerely, 

The authors

Editor

Dear Dr. ...,

Thank you for submitting your manuscript to PLOS ONE. After careful consideration, we feel that it has merit but does not fully meet PLOS ONE’s publication criteria as it currently stands. Therefore, we invite you to submit a revised version of the manuscript that addresses the points raised during the review process.

• A rebuttal letter that responds to each point raised by the academic editor and review-er(s). You should upload this letter as a separate file labeled 'Response to Reviewers'.

• An unmarked version of your revised paper without tracked changes. You should up-load this as a separate file labeled 'Manuscript'.

If applicable, we recommend that you deposit your laboratory protocols in protocols.io to enhance the reproducibility of your results. Protocols.io assigns your protocol its own identi-fier (DOI) so that it can be cited independently in the future. For instructions see: https://journals.plos.org/plosone/s/submission-guidelines#loc-laboratory-protocols. Ad-ditionally, PLOS ONE offers an option for publishing peer-reviewed Lab Protocol articles, which describe protocols hosted on protocols.io. Read more information on sharing protocols at https://plos.org/protocols?utm_medium=editorial-email&utm_source=authorletters&utm_campaign=protocols.

We look forward to receiving your revised manuscript.

Kind regards,

Pedro Ribeiro Mucharreira, Ph.D.

Academic Editor

PLOS ONE

Journal Requirements:

2. In your Data Availability statement, you have not specified where the minimal data set un-derlying the results described in your manuscript can be found. PLOS defines a study's min-imal data set as the underlying data used to reach the conclusions drawn in the manuscript and any additional data required to replicate the reported study findings in their entirety. All PLOS journals require that the minimal data set be made fully available. For more information about our data policy, please see http://journals.plos.org/plosone/s/data-availability.

"Upon re-submitting your revised manuscript, please upload your study’s minimal underly-ing data set as either Supporting Information files or to a stable, public repository and include the relevant URLs, DOIs, or accession numbers within your revised cover letter. For a list of acceptable repositories, please see http://journals.plos.org/plosone/s/data-availability#loc-recommended-repositories. Any potentially identifying patient information must be fully anonymized.

Important: If there are ethical or legal restrictions to sharing your data publicly, please explain these restrictions in detail. Please see our guidelines for more information on what we consid-er unacceptable restrictions to publicly sharing data: http://journals.plos.org/plosone/s/data-availability#loc-unacceptable-data-access-restrictions. Note that it is not acceptable for the authors to be the sole named individuals responsible for ensuring data access.

Thank you for pointing toward the items for submitting our revised manuscript as well as to the Journal Requirements below. We have carefully addressed all remarks and revised the manuscript accordingly and submitted the present rebuttal letter, a marked-up copy of our manuscript as well as an unmarked version without tracked changes and labeled them ac-cordingly. 

We believe that by taking your and reviewers’ feedback into account, the paper has been im-proved, and we hope that it can now be accepted for publication.

Journal Requirements:

2. In your Data Availability statement, you have not specified where the minimal data set un-derlying the results described in your manuscript can be found. PLOS defines a study's min-imal data set as the underlying data used to reach the conclusions drawn in the manuscript and any additional data required to replicate the reported study findings in their entirety. All PLOS journals require that the minimal data set be made fully available. For more information about our data policy, please see http://journals.plos.org/plosone/s/data-availability.

"Upon re-submitting your revised manuscript, please upload your study’s minimal underly-ing data set as either Supporting Information files or to a stable, public repository and include the relevant URLs, DOIs, or accession numbers within your revised cover letter. For a list of acceptable repositories, please see http://journals.plos.org/plosone/s/data-availability#loc-recommended-repositories. Any potentially identifying patient information must be fully anonymized.

Important: If there are ethical or legal restrictions to sharing your data publicly, please explain these restrictions in detail. Please see our guidelines for more information on what we consid-er unacceptable restrictions to publicly sharing data: http://journals.plos.org/plosone/s/data-availability#loc-unacceptable-data-access-restrictions. Note that it is not acceptable for the authors to be the sole named individuals responsible for ensuring data access.

Thank you for pointing out the Journal Requirements!

First, we have prepared the manuscript according to the PLOS ONE style template.

Second, we uploaded our Minimal Dataset and Codebook to a public repository here (https://osf.io/vwd73/?view_only=6992cf4dc16b4401b9cbe7006683002e). Thank you for updating the Data Availability statement for us. 

Third, we double-checked our reference list to insure its correctness.

 

Reviewers’ comments

Thank you very much for your helpful comments!

(1) 1. Is the manuscript technically sound, and do the data support the conclu-sions?

Reviewer #1: Yes

Reviewer #2: Partly

Reviewer #3: Yes

Reviewer #4: Yes

 Thank you for pointing us to further explain our technical approach. To clarify our data structure, we added clear descriptions in the methods section, to make our ap-proach easier to grasp. Further, we have updated and refined Fig. 2 and the table of Covariances, Tables S4, with additional explanations. We hope this clarifies how we handled our data to support our conclusion.

Page 19: “In the first months of Grades 1 and 3, students’ mathematics and language achievement scores were collected with standardized achievement tests. Expert groups consisting of teachers, content specialists in teaching and learning, and psychometri-cians developed these tests to ensure the content validity of these tests [1], based on the Luxembourgish national curriculum standards (1). On the day of testing, the stu-dents completed the achievement tests in their own classrooms in a paper-and-pencil format.”

Page 20: “Table 2 shows these reliability coefficients for both mathematics and lan-guage in Grades 1 and 3 in 2015-2017 and 2017-2019. [...] The students answered items from three domains of mathematics competence: “numbers and operations,” “space and shape,” and “size and measurement” (see, for a more comprehensive ex-planation https://epstan.lu/en/assessed-competences-21/). In Grade 3, the students took the mathematics tests in German because the students had been taught in German during Grades 1 and 2. Again, the students answered items from three mathematics competence domains: “numbers and operations”, “space and form”, and the novel area of “quantities and measures” (see, for a more comprehensive explanation https://epstan.lu/en/assessed-competences-31/).”

Page 20: “We averaged the scores for listening and reading comprehension as prior language achievement in the VA model to have one single score for language achieve-ment. [...] In Grade 3, the students took the listening and reading comprehension lan-guage achievement tests in German, which had been the language of instruction dur-ing Grades 1 and 2.”

Page 24: “We estimated the random effects within each school to obtain VA scores via Equations 1 and 2 [3]. More specifically, these were the Level 1 residuals from the multilevel model, averaged within a school. In other words, all student-level VA scores were averaged into one VA score per school. […] Prior achievement in mathe-matics and in language, SES, migration status, language spoken at home, and sex are covariates, thus being statistically accounted for.”

Page 25: “We defined schools with a stable VA score as schools that remained in the same VA rank quartile in 2017 and 2019. We defined schools with an unstable VA score as schools that were in different quartiles in 2017 and 2019.”

Page 25: “Fig 2. Graphical Representation of the VA Score Intercorrelations across Time and Domains. Double-headed arrows signify the correlations we calcu-lated.

 ”

Supplemental Material, S4 Table: “S4 Table. Covariance Table of VA Scores with Different Outcome Domains (Mathematics and Language) over Time (2017 to 2019).

Note. Covariances are calculated on the basis of n = 7,016 elementary school students’ VA scores. The VA scores of 2017 and 2019 were informed by data from 2015 and 2017, respectively.”

(2) 2. Has the statistical analysis been performed appropriately and rigorously? 

Reviewer #1: Yes

Reviewer #2: No

Reviewer #3: Yes

Reviewer #4: Yes

 We thank all reviewers for their assessment and are again grateful for their input. We have carefully re-read our statistical analysis section and clarified formerly possible hard-to-understand passages for greater clarity. 

Page 24: “We estimated the random effects within each school to obtain VA scores via Equations 1 and 2 [3]. More specifically, these were the Level 1 residuals from the multilevel model, averaged within a school. In other words, all student-level VA scores were averaged into one VA score per school. […] Prior achievement in mathe-matics and in language, SES, migration status, language spoken at home, and sex are covariates, thus being statistically accounted for.”

Page 25: “We defined schools with a stable VA score as schools that remained in the same VA rank quartile in 2017 and 2019. We defined schools with an unstable VA score as schools that were in different quartiles in 2017 and 2019.”

Page 25: “Fig 2. Graphical Representation of the VA Score Intercorrelations across Time and Domains. Double-headed arrows signify the correlations we calcu-lated.

 ”

We hope and trust that these changes improved the clarity of our statistical analyses.

(3) 3. Have the authors made all data underlying the findings in their manuscript fully available?

The PLOS Data policy requires authors to make all data underlying the findings de-scribed in their manuscript fully available without restriction, with rare exception (please refer to the Data Availability Statement in the manuscript PDF file). The data should be provided as part of the manuscript or its supporting information, or deposit-ed to a public repository. For example, in addition to summary statistics, the data points behind means, medians and variance measures should be available. If there are restrictions on publicly sharing data—e.g. participant privacy or use of data from a third party—those must be specified.

Reviewer #1: Yes

Reviewer #2: No

Reviewer #3: Yes

Reviewer #4: Yes

 We thank you for this important remark. As stated in our Data Availability statement, we now make a Minimal Dataset and Codebook of School’s VA Quartile Ranking Across Time and Domains available in our Supplemental Material. Together with the statistical syntax, the readers can track and understand our statistical procedure.

Please, find the online version of the Minimal Dataset and Codebook here: https://osf.io/vwd73/?view_only=6992cf4dc16b4401b9cbe7006683002e

Page 28: “The S3 Syntax shows the R code we used to analyze the correlations be-tween the VA scores. The S4 Table shows the covariance table of VA scores with dif-ferent outcome domains (mathematics and language) over time. The S5 Dataset shows the minimal dataset and codebook of school’s VA quartile ranking across time and domains.”

Supplemental Material: “S5 Dataset. Minimal Dataset and Codebook of School’s VA Quartile Ranking Across Time and Domains.”

We appreciate your support in enhancing the replicability of our manuscript by hold-ing us to the highest standards of the Open Science movement.

(4) 4. Is the manuscript presented in an intelligible fashion and written in standard English?

Reviewer #1: Yes

Reviewer #2: No

Reviewer #3: Yes

Reviewer #4: Yes

 Thank you for pointing out the quality of our writing or the way there! We have had the manuscript proofread by a professional proofreader who is an English native speaker. We are happy to correct any further errors the reviewers point out. 

Thank you in advance for any clarification or specific issues we can improve con-cerning our writing.

(5) 5. Review 1‘s Comments to the Author

Reviewer #1: The paper is a high-quality piece of research on the Value-added by a teacher or a school to their students’ achievement. It is based on a structured explora-tion of relevant literature, as well as on a rigorous empirical assessment, providing insightful results that are aligned with previous findings. Nevertheless, before publica-tion, in the conclusions it is recommended to provide implications at the public policy level, that is, education policy including the supply and demand for both public and private education services.

 Thank you very much for kind and constructive assessment of our study. We were happy to rewrite the conclusions sections to include more implications for public pol-icy.

Page 30: “Given our correlational findings and the multiple sources of instability, we conclude that VA scores have insufficient predictive power for the value a school adds to its students over time. Thus, these scores cannot be considered stable enough over time to be used for high-stakes decisions [4,5]. As VA scores are a highly political topic and their use is controversially discussed, they should not be used as the sole indicator for a schools effectiveness on a public policy level [6,7]. In the public inter-est, however, schools with stable VA scores can be used in research contexts for in-formative purposes, including learning from schools with stable high VA scores.”

Pages 30 - 31: “Especially for the 10 schools that changed from the highest to the lowest VA ranks or vice versa, this instability could have dramatic consequences if VA scores were applied for accountability reasons, such as decisions about funding or the closing of schools. With no major advances in the stability of VA scores by adding richer sets of background variables or more advanced statistical methods, on its own, “Value-added is [still] of little value” [8] to public educational policy.”

Page 33: “The present study provided evidence for the moderate stability of primary schools’ VA scores. Only 34-38% of the schools showed stable VA scores across two years with moderate correlations of r = .34 for mathematics and r = .37 for language achievement as the outcome domains. The number of stable schools did not differ greatly between mathematics and language. Real-life implications for schools may be consequential with only about one third of schools having a stable VA score over time. This finding indicates that VA scores should not be used as the only measure for pur-poses of accountability. Thus, both public and private educational services should re-frain from using VA scores as the (sole) metric to rank schools in their effectiveness and policymakers need to consider the present controversy about school VA scores. Complementary sources of data to make appropriate educational decisions are strongly recommended, as school VA scores do not seem to be stable enough over time and theoretical assumptions about the VA scores don’t seem to hold in practice [7,9].”

Thank you for your kind recommendation. We are optimistic, we could address it and feel it has much improved the overall message of our article. We hope for your ap-proval.

(6) 6. Review 2‘s Comments to the Author

Reviewer #2: The title "Value-Added Scores Show Limited Stability over Time in Primary School" addresses a topic of extreme interest, but difficult to read, with imag-es that do not help to clarify the theory it presents.

The article with profound changes has potential for future publication, at the moment it seems unclear to me.

a) Poor introdution

b) Insufficient framing, I suggest 

I. updating the literature review 

II. and improving the specification of models. 

III. Presenting a table of the variables used in studies of the same genre and their respective conclusions, helps to better understand the frame-work and later the discussion and conclusion.

c) Correct numeric values, for example, .15 means 0.15.

d) Develop the method and improve the explanation of academic performance.

e) Weak Data analysis, discussion and conclusion. It is unclear whether the data are used year by year, or pooled for the 3 years under study, namely in calcu-lating the Covariance Table of VA Scores with Different Outcome Domains (Math and Language) over time.

Good luck for work development!

 We thank you for your important remarks and made several changes to improve our manuscript accordingly. These changes are outlined below in greater detail.

a) We have revised the introduction section and made several smaller changes. However, we are open to more specific remarks, how we can improve the in-troduction of our manuscript.

Page 3: “The VA score quantifies the difference between the expected achievement of students with similar background characteristics and their actual achievement [10]. Positive VA scores signify higher-than-expected achievement, given the student’s background characteristics (e.g. socioeconomic status [SES], language, or prior achievement), whereas negative scores imply lower-than-expected achievement. At-tempting to make a fair comparison between schools, these student VA scores can be averaged per school (or teacher) and indicate the value a school adds to its students [11,12] [deleted] independent of their background.”

b) We have worked to improve the framing of our study.

I. We updated our literature review to ensure more recent literature is as well represented as older seminal works. Please, find a few exam-ples here:

Page 8: “Despite this moderate to large stability but given the dependence on student characteristics, Perry [5] and more recent research from the UK [13] recommended avoiding school VA scores as a basis for policymaking or other high-stakes deci-sions.”

Page 14: “This diversity leads to different preconditions for learning mathematics and new languages (or even the language of instruction) and thus shapes students’ school careers [14] and school completion [15].”

Page 15: “Otherwise, VA scores would fail to flexibly adjust to the constantly evolv-ing language and school landscape (1) and should thereby not be used in high-stakes decision making.”

II. We expanded our explanation of the model specification and added some more explanatory sentences.

Page 24: “We estimated the random effects within each school to obtain VA scores via Equations 1 and 2 [3]. More specifically, these were the Level 1 residuals from the multilevel model, averaged within a school. In other words, all student-level VA scores were averaged into one VA score per school. […] Prior achievement in mathe-matics and in language, SES, migration status, language spoken at home, and sex are covariates, thus being statistically accounted for.”

Page 25: “We defined schools with a stable VA score as schools that remained in the same VA rank quartile in 2017 and 2019. We defined schools with an unstable VA score as schools that were in different quartiles in 2017 and 2019.”

III. Thank you very much for this valuable remark. We included a table of the studies on value-added modeling across time with their respective included variables, samples, and conclusions. As suggested by you, this gives a much better overview on the literature in this field at one glance and renders our manuscript much easier to understand. Fur-ther, we referred to recent and comprehensive research on the use of variables and model specifications in value-added research [16,17]. As a systematic review specifically on the stability of VA scores over time would have been outside the scope of our present manuscript, we encouraged the readers to follow this idea by Reviewer 2. We strongly believe such an overview would be a worthwhile endeavor and would advance the field of VA scores both methodologically and theoretically. 

Page 8: “Research on school-level stability over time is still scarce and has produced mixed results. Table 1 gives an overview of prior research on the stability of school VA scores over time with the included variables and samples.”

Page 9: “Table 1. Overview of Prior Research on the Stability of School Value-Added Scores Over Time, their Included Variables, Samples, and Conclusions.

Note. n = number of schools in the sample; SES = socio-economic status; SEN = Special Educational Needs; VA = Value-Added; IC = students who have been ‘In Care’ at any time while being at this school; IDACI = Income Deprivation Affecting Children Index measuring deprivation based on student postcode. Please see the re-spective original study for more details.”

Page 32-33: “Future research could tackle the stability of VA scores and the implica-tions of stability in these scores in three distinct ways: (a) by investigating fundamen-tal research questions on the parameters that influence the stability of VA scores, (b) by conducting a systematic review of both published and gray literature on the stabil-ity of VA scores over time, and (c) by looking at the real-life implications that (un)stable VA scores have for schools.”

c) We double-checked that all numeric values, which can be larger than 1 in-clude a 0 before the decimal separator. For correlations, however, we did not include a zero before the decimal separators, as they cannot be larger than 1. This is in line with common reporting guidelines in our field [18] and also seems to be common practice in articles from other fields in PLOS ONE [19].

d) We developed the method further (see our reply to your comment II) and elaborated further on the explanation of academic achievement measures.

Page 19: “In the first months of Grades 1 and 3, students’ mathematics and language achievement scores were collected with standardized achievement tests. Expert groups consisting of teachers, content specialists in teaching and learning, and psychometri-cians developed these tests to ensure the content validity of these tests [1], based on the Luxembourgish national curriculum standards (1). On the day of testing, the stu-dents completed the achievement tests in their own classrooms in a paper-and-pencil format.”

Page 20: “Table 2 shows these reliability coefficients for both mathematics and lan-guage in Grades 1 and 3 in 2015-2017 and 2017-2019. [...] The students answered items from three domains of mathematics competence: “numbers and operations,” “space and shape,” and “size and measurement” (see, for a more comprehensive ex-planation https://epstan.lu/en/assessed-competences-21/). In Grade 3, the students took the mathematics tests in German because the students had been taught in German during Grades 1 and 2. Again, the students answered items from three mathematics competence domains: “numbers and operations”, “space and form”, and the novel area of “quantities and measures” (see, for a more comprehensive explanation https://epstan.lu/en/assessed-competences-31/).”

Page 20: “We averaged the scores for listening and reading comprehension as prior language achievement in the VA model to have one single score for language achieve-ment. [...] In Grade 3, the students took the listening and reading comprehension lan-guage achievement tests in German, which had been the language of instruction dur-ing Grades 1 and 2.”

e) We carefully read through the explanation of our data analysis, discussion, and conclusion and made improvements. We hope that the data analysis has become clearer with the improvement of figures, tables, and some illustrative examples as detailed above. In line with your and Reviewer 4’s comments, we also worked on our discussion and conclusion to put a stronger focus on the educational policy implications of our findings.

Page 30: “Given our correlational findings and the multiple sources of instability, we conclude that VA scores have insufficient predictive power for the value a school adds to its students over time. Thus, these scores cannot be considered stable enough over time to be used for high-stakes decisions [4,5]. As VA scores are a highly political topic and their use is controversially discussed, they should not be used as the sole indicator for a schools effectiveness on a public policy level [6,7]. In the public inter-est, however, schools with stable VA scores can be used in research contexts for in-formative purposes, including learning from schools with stable high VA scores.”

Pages 30 - 31: “Especially for the 10 schools that changed from the highest to the lowest VA ranks or vice versa, this instability could have dramatic consequences if VA scores were applied for accountability reasons, such as decisions about funding or the closing of schools. With no major advances in the stability of VA scores by adding richer sets of background variables or more advanced statistical methods, on its own, “Value-added is [still] of little value” [8] to public educational policy.”

Page 33: “The present study provided evidence for the moderate stability of primary schools’ VA scores. Only 34-38% of the schools showed stable VA scores across two years with moderate correlations of r = .34 for mathematics and r = .37 for language achievement as the outcome domains. The number of stable schools did not differ greatly between mathematics and language. Real-life implications for schools may be consequential with only about one third of schools having a stable VA score over time. This finding indicates that VA scores should not be used as the only measure for pur-poses of accountability. Thus, both public and private educational services should re-frain from using VA scores as the (sole) metric to rank schools in their effectiveness and policymakers need to consider the present controversy about school VA scores. Complementary sources of data to make appropriate educational decisions are strongly recommended, as school VA scores do not seem to be stable enough over time and theoretical assumptions about the VA scores don’t seem to hold in practice [7,9].”

To clarify how we pooled the data to calculate the covariances over time and across domains, we complemented the covariance matrix in our supplemental table S4 with a short explanation.

Supplemental Material, S4 Table: “S4 Table. Covariance Table of VA Scores with Different Outcome Domains (Mathematics and Language) over Time (2017 to 2019).

Note. Covariances are calculated on the basis of n = 7,016 elementary school students’ VA scores. The VA scores of 2017 and 2019 were informed by data from 2015 and 2017, respectively.”

Additional to the changes in the text and the Supplemental Material S4, we have im-proved Fig. 2 to explain more precisely how we used the data over time and across outcome domains.

Page 25: “Fig 2. Graphical Representation of the VA Score Intercorrelations across Time and Domains. Double-headed arrows signify the correlations we calcu-lated.

 ”

We hope that these changes have allayed your concerns in our manuscript and have improved its clarity. We appreciate your support in enhancing the consistency of our manuscript and thank you for your comments.

(7) 7. Review 3‘s Comments to the Author

Reviewer #3: The statistical analysis seems to be very well integrated within the con-text, and the article is very thoroughly documented and well written. I recommend the article to be published as it is, and I consider it is a very useful contribution to the journal.

 We thank you very much for your kind approval of our manuscript and appreciate your support.

(8) 8. Review 4‘s Comments to the Author

Reviewer #4: It seems to me, in my opinion, that the article focuses on a topic of great interest and relevance. It is reasonably well written, and provides a list of adequate references, both in number and current.

Maybe the images could be improved a little more, in terms of quality, but overall it seems like a good article.

 We thank you for this important remark and made several changes to the images. Overall, we have improved the quality of all images and ensured a high resolution. Please, see the specific changes we made below. In all images, we have kept the re-spective overall color scheme as these colors can be differentiated by color blind readers (we checked for eight of the most common kinds of color vision deficiency).

In Fig. 1, we added the tick marks on the x-axis for clarity, muted the colors for a nicer look, added an explanation to the “expected achievement” line, and optimized the fond. 

Page 25: “Fig 1. Illustration of high and low VA schools performing above and below what is expected of them from one measurement point to another. Dou-ble-headed arrows signify the VA score as the difference between a school’s expected achievement and its actual achievement.

 ”

In Fig. 2, we decluttered the ovals and added information on the type of correlation next to the double-headed arrows. As we had removed the year of data collection from the ovals, we now display it at the top for greater clarity. In line with on of Re-viewer 2’s comments, we added a note, explaining the relevance of the prior achieve-ment data two years earlier

Page 25: “Fig 2. Graphical Representation of the VA Score Intercorrelations across Time and Domains. Double-headed arrows signify the correlations we calcu-lated.

 ”

In Fig. 3, we muted the colors for better readability of the numbers but did not change the overall color scheme, to keep the notion of red signifying low VA and green signifying high VA.

Page 27: “Fig 3. Transition Diagram Indicating Changes in the Schools’ Value-Added Score Ranking Quartiles from 2017 to 2019. A school’s position in the VA score ranking is indicated on either side of the transition diagram. High VA scores are in the top 25%, Upper medium VA scores are between the 50th and 75th percentiles, Lower medium VA scores are between the 25th and 50th percentiles, and low VA scores are in the lowest 25%.

 ”

Thank you again for your help in improving our manuscript.

(9) PLOS authors have the option to publish the peer review history of their article (what does this mean?). If published, this will include your full peer review and any attached files.

Do you want your identity to be public for this peer review? For information about this choice, including consent withdrawal, please see our Privacy Policy.

Reviewer #1: Yes: João Leitão

Reviewer #2: No

Reviewer #3: No

Reviewer #4: No

 Thank you very much to all reviewers for their feedback. Pleased to meet you, Profes-sor João Leitão, and thank you.

 

References

 1. Fischbach A, Ugen S, Martin R. ÉpStan Technical Report. Luxembourg: University of Luxembourg; 2014. Available: http://hdl.handle.net/10993/15802

2. Ministry of National Education, Children and Youth. Elementary School. Cycles 1 -4. The Levels of Competence. 2011. Available: http://www.men.public.lu/catalogue-publications/fondamental/apprentissages/documents-obligatoires/niveaux-competences/en.pdf

3. Ferrão ME, Goldstein H. Adjusting for measurement error in the value added model: evi-dence from Portugal. Qual Quant. 2009;43: 951–963. doi:10.1007/s11135-008-9171-1

4. Gorard S, Hordosy R, Siddiqui N. How unstable are “school effects” assessed by a value-added technique? International Education Studies. 2013;6: 1–9. doi:10.5539/ies.v6n1p1

5. Perry T. English value-added measures: Examining the limitations of school performance measurement. Br Educ Res J. 2016;42: 1056–1080. doi:10.1002/berj.3247

6. Amrein-Beardsley A, Holloway J. Value-Added Models for Teacher Evaluation and Ac-countability: Commonsense Assumptions. Educational Policy. 2019;33: 516–542. doi:10.1177/0895904817719519

7. Conaway C, Goldhaber D. Appropriate Standards of Evidence for Education Policy Deci-sion Making. Education Finance and Policy. 2020;15: 383–396. doi:10.1162/edfp_a_00301

8. Gorard S. Value‐added is of little value. Journal of Education Policy. 2006;21: 235–243. doi:10.1080/02680930500500435

9. Scherrer J. Measuring teaching using value-added modeling: The imperfect panacea. NASSP Bulletin. 2011;95: 122–140. doi:10.1177/0192636511410052

10. Sanders WL, Wright SP, Horn SP. Teacher and classroom context effects on student achievement: Implications for teacher evaluation. Journal of Personnel Evaluation in Edu-cation. 1997;11: 57–67. doi:10.1023/A:1007999204543

11. Tymms P. Baseline assessment, value-added and the prediction of reading. Journal of Re-search in Reading. 1999;22: 27–36. doi:10.1111/1467-9817.00066

12. Braun H. Using Student Progress to Evaluate Teachers: A Primer on Value-Added Mod-els. Educational Testing Service: Educational Testing Service; 2005. 

13. Leckie G, Goldstein H. The importance of adjusting for pupil background in school value‐added models: A study of Progress 8 and school accountability in England. Br Educ Res J. 2019;45: 518–537. doi:10.1002/berj.3511

14. Hadjar A, Backes S. Bildungsungleichheiten am Übergang in die Sekundarschule in Lu-xemburg. 2021 [cited 13 Apr 2022]. doi:10.48746/BB2021LU-DE-21A

15. Ferrão ME. The evaluation of students’ progression in lower secondary education in Bra-zil: Exploring the path for equity. Studies in Educational Evaluation. 2022;75: 101220. doi:10.1016/j.stueduc.2022.101220

16. Levy J, Brunner M, Keller U, Fischbach A. Methodological issues in value-added model-ing: an international review from 26 countries. Educ Asse Eval Acc. 2019;31: 257–287. doi:10.1007/s11092-019-09303-w

17. Levy J, Brunner M, Keller U, Fischbach A. How sensitive are the evaluations of a school’s effectiveness to the selection of covariates in the applied value-added model? Educ Asse Eval Acc. 2022 [cited 25 May 2022]. doi:10.1007/s11092-022-09386-y

18. American Psychological Association (Washington, District of Columbia), editor. Publica-tion manual of the American psychological association. Seventh edition. Washington, DC: American Psychological Association; 2020. 

19. Genschow O, van Den Bossche S, Cracco E, Bardi L, Rigoni D, Brass M. Mimicry and automatic imitation are not correlated. Iacoboni M, editor. PLoS ONE. 2017;12: e0183784. doi:10.1371/journal.pone.0183784

---

## [Editor Report · Decision Letter 1]

4 Dec 2022

Value-Added Scores Show Limited Stability over Time in Primary School

PONE-D-22-22428R1

Dear Dr. Valentin Emslander

We’re pleased to inform you that your manuscript has been judged scientifically suitable for publication and will be formally accepted for publication once it meets all outstanding technical requirements.

Kind regards,

Pedro Ribeiro Mucharreira, Ph.D.

Academic Editor

PLOS ONE
---

## [Editor Report · Acceptance letter]

14 Dec 2022

PONE-D-22-22428R1 

Value-Added Scores Show Limited Stability over Time in Primary School 

Dear Dr. Emslander:

I'm pleased to inform you that your manuscript has been deemed suitable for publication in PLOS ONE. Congratulations! Your manuscript is now with our production department. 

Kind regards, 

on behalf of

Dr. Pedro Ribeiro Mucharreira 

Academic Editor

PLOS ONE